# Oral-Health-Related Quality of Life and Anxiety in Orthodontic Patients with Conventional Brackets

**DOI:** 10.3390/ijerph191710767

**Published:** 2022-08-29

**Authors:** Adrián Curto, Alejandro Alvarado-Lorenzo, Alberto Albaladejo, Alfonso Alvarado-Lorenzo

**Affiliations:** 1Department of Surgery, Pediatric Dentistry, Faculty of Medicine, University of Salamanca, Avenida Alfonso X El Sabio s/n, 37007 Salamanca, Spain; 2Department of Surgery, Orthodontics, Faculty of Medicine, University of Salamanca, Avenida Alfonso X El Sabio s/n, 37007 Salamanca, Spain

**Keywords:** orthodontics, anxiety, STAI, oral-health-related quality of life, OHIP-14, self-esteem

## Abstract

The aim of this study was to evaluate the oral-health-related quality of life (OHRQoL) and anxiety levels of adult patients undergoing orthodontic treatment with fixed multibracket appliances. Materials and Methods: The study was carried out at the Dental Clinic of the University of Salamanca in 2021. It included 120 adult patients between 19 and 45 years of age undergoing orthodontic treatment with conventional metal brackets. The data collection instruments chosen were the state–trait anxiety inventory (STAI) to assess anxiety levels and the OHIP-14 questionnaire to measure the OHRQoL. Anxiety levels and OHRQoL were analyzed one month after starting treatment. Results: The mean age was 31.7 years ± 6.5 years; 68 patients were women (56.7%) and 52 were men (43.3%). Psychological disability was the dimension of the OHIP-14 questionnaire that was found to have the greatest impact (3.20 ± 1.08) on patients, as compared to the dimension of disability, which had the lowest impact on the oral-health quality of life (0.37 ± 0.56). The mean total score of the OHIP-14 questionnaire was 11.93 (±2.19). There was no statistically significant influence from either sex or age on the anxiety and oral-health quality of life of the participants; however, there was a significant relationship between the dimensions of physical disability and anxiety traits. Conclusions: The physical disability dimension of the OHIP-14 questionnaire increased the anxiety level of adult patients treated with conventional brackets. The impact of orthodontic treatment on adult patients may negatively influence their levels of anxiety.

## 1. Introduction

Improvement in facial appearance and obtaining a correct occlusion are the main goals in orthodontic treatment. Awareness of how facial appearance affects the oral-health quality of life has, in recent years, increased the demand for orthodontic treatment in the adult population [1,2].

Pain is a common symptom in patients during the early stages of treatment. In addition to the physical discomfort that patients describe during the early stages of their orthodontic treatment, psychological aspects, such as anxiety, may be affected during treatment [3]. Dental anxiety and fear are the most common traits found in most patients who visit the dental clinic. Dental anxiety is observed with greater prevalence in female adult patients [4,5]. It has been observed that patients with higher levels of anxiety during orthodontic treatment will also describe higher levels of pain [6]. The presence of dental anxiety can hinder social and daily activities, especially eating and sleeping [7].

The implementation of strategies to reduce the level of anxiety in orthodontic patients can help to decrease their perception of pain. Dental anxiety cannot be completely eliminated in patients despite all the advances in the field of dentistry [8]. Tooth anxiety has a negative impact on patients’ oral-health quality of life [9].

It is now recognized that health contributes to an improvement in the health-related quality of life (HRQoL), which is now recognized as an important parameter for patient assessment in most health care fields, including orthodontics. Oral-health-related quality of life (OHRQoL) has important implications for patient dental care [10]. Several studies have analyzed the psychological and social effects of orthodontic treatments on the OHRQoL [11,12,13,14,15]. Facial aesthetics, dental aesthetics, and functional occlusion are different factors that influence the motivation of patients to undergo orthodontic treatment and, consequently, the decrease in the psychosocial problems that influence the OHRQoL [16]. The OHRQoL can be considered a useful tool in the planning and evolution of orthodontic treatment and in the evaluation of the objectives achieved [13]. According to the scientific literature, the phase of orthodontic treatment may either worsen or improve the impact that treatment has on the OHRQoL of patients [15]. At the end of orthodontic treatment, patients describe a significant improvement in their OHRQoL compared to at the start of treatment [15,17,18]. As a result of the review of the scientific literature, it is believed that there may be a relationship between dental anxiety and the OHRQoL [19]. During orthodontic treatment, the severity of the malocclusion can influence the degree of compliance in patients. The cooperation of patients during their treatment is essential in order to achieve the treatment goals [20].

Despite the potential association between the level of dental anxiety and the level of pain described by the patients, there are no published studies that evaluate the levels of anxiety and the impact on the OHRQoL in adult patients during the first stage of their orthodontic treatment with brackets that also analyze the influence the sex and the age of the patients has on these two variables.

The aim of this study was to evaluate the level of anxiety and the OHRQoL of adult patients during the initial stage of their orthodontic treatment using fixed multibracket appliances. The influence of sex and age on the OHRQoL and on anxiety was also analyzed.

Therefore, the null hypothesis of our study was that there was no correlation between patients with fixed multibracket appliances and anxiety and the OHRQoL.

## 2. Materials and Methods

### 2.1. Study Design

This study was approved by the Bioethics Committee of the University of Salamanca (USAL 142/20). The study followed the guidelines of the Helsinki Declaration and was designed according to the STROBE guidelines. Participants were informed about the review procedure and were assured of the confidentiality of the information collected. All study participants signed an informed consent prior to inclusion in the study.

Participants were recruited by selecting all patients who began orthodontic treatment using fixed brackets between January and May 2021 at the Dental Clinic of the University of Salamanca.

Bracket bonding, arch insertion, and orthodontic treatment were performed by the same experienced technician. Conventional metal brackets (Victory Series^®^, 3M, St. Paul, MN, USA) were used.

The inclusion criteria were as follows: (1) adult patients over 18 years of age; (2) patients in need of orthodontic treatment; (3) patients eligible for treatment with fixed multibrackets; and (4) dental crowding between −2 and −6 mm.

The exclusion criteria were as follows: (1) patients with previous orthodontic treatment; (2) patients in need of extractions during their orthodontic treatment; (3) dental caries and/or untreated periodontal pathology; (4) patients in need of orthodontic surgical treatment; (5) patients with craniofacial abnormalities such as cleft lip or palate; (6) pregnant patients; and (7) patients previously diagnosed with anxiety and/or on pharmacological treatment.

### 2.2. Anxiety Analysis

The level of patient anxiety was assessed using the state–trait anxiety inventory (STAI). The STAI questionnaire is a 40-item Likert scale that evaluates separate dimensions of anxiety as a state (items 1–20) and anxiety as a trait (items 21–40). In total, 17 of these items must be recoded (reverse score) before calculating the total questionnaire score. In general, a score greater than 40 points is considered to be an indication of a high degree of anxiety [21]. This questionnaire was provided to patients one month after starting their orthodontic treatment. The STAI anxiety questionnaire has been used as a tool to determine the level of dental anxiety in orthodontic patients in previous studies [22,23].

### 2.3. Oral-Health-Related Quality of Life Analysis

The OHRQoL was analyzed using the OHIP-14 questionnaire one month after starting treatment. The OHRQoL questionnaire OHIP-14 (Oral Health Impact Profile-14) has seven domains; each of the domains is made up of two items, which gives rise to a total of fourteen questions. The seven domains are functional limitation, physical pain, psychological discomfort, physical disability, psychological disability, social disability, and handicap. Each question is scored on a five-point Likert scale, ranging from 0 (never) to 4 (always), and the final score is the sum of all the individual items. The total score of the OHIP-14 questionnaire ranges from 0 to 56, the higher scores indicating a worse impact on patients’ OHRQoL [24]. The OHIP-14 questionnaire has been used in numerous studies to assess the impact of orthodontic treatment on patients’ OHRQoL [25]. In this study, the Spanish version of the OHIP-14 questionnaire was used, which has been previously validated [26].

### 2.4. Evaluation of Sample Size

Based on data from previously published studies that have also evaluated oral-health quality of life [25], the primary endpoint (total OHIP-14 score) has a standard deviation ranging from 6 to 10 points. Considering the median point between these values, a minimum sample size of 116 participants was estimated. Finally, 120 participants were recruited who answered all the questions in the questionnaires.

### 2.5. Statistical Analysis

The statistical analysis was carried out using IBM-SPSS Statistics software version 25 (IBM, Armonk, NY, USA). The statistical techniques and tests used were: 1. frequency and percentage tables for qualitative variables; 2. measures of centrality (mean and median) and variability (observed range, standard deviation, and interquartile amplitude) for quantitative variables; 3. for the contrast between the means of groups of different subjects (independent of each other), Student’s t-test was used when the variables were normal, and the nonparametric alternative (Mann–Whitney) was used when they were not; and 4. for correlations between the variables, the Spearman coefficient was used.

In all these statistical tests, significance was considered to be when *p* < 0.05 and high significance to be when *p <* 0.01.

## 3. Results

### 3.1. Characteristics of Participants

Of the 120 patients, 68 patients were female (56.7%), and 52 were male (43.3%). The ages of the participants in the study ranged from 19 to 45 years, with a mean age of 31.7 years (±6.5 years). No statistically significant differences were observed between the ages of men and women (*p* = 0.443).

### 3.2. Anxiety Levels

The anxiety values were concentrated in the middle-low part of the scale without any participant in the study exceeding 35 points. The average values were around 27–28 points; therefore, it can be concluded that this study group did not present with anxiety problems.

When analyzing anxiety levels within the anxiety states, patients described higher scores compared to the anxiety traits (Table 1).

### 3.3. Oral-Health-Related Quality of Life

From the average values found when analyzing the OHRQoL of the participants in the study, we concluded that the sample studied, in general, possessed a good oral-health quality of life, especially in the dimensions of physical disability, social disability, and handicap as described in the OHIP-14 questionnaire (with averages that were less than one).

The dimension of the OHIP-14 questionnaire that had the greatest impact was psychological disability (3.20 ± 1.08), compared to the handicap dimension, which had the lowest score in the questionnaire (0.37 ± 0.56). The mean total score of the OHIP-14 questionnaire was 11.93 (±2.19) (Table 2 and Figure 1).

### 3.4. Influence of Sex on OHRQoL and Anxiety Levels

The results shown in Table 3 indicate that there were no variables with statistically significant differences. We can conclude that no significant evidence was found to deduce that a lower impact in the OHRQoL or a lower level of anxiety in the study participants could be differentiated by sex.

### 3.5. Influence of Age on OHRQoL and Anxiety Levels

To evaluate the influence of age on the OHRQoL of the study participants and their anxiety levels, the group was categorized by this factor. The cutoff point was considered to be 30 years, a value close to the median, generating two groups with a similar sample size. The averages of these two groups were compared with the measured variables. The results described in Table 4 indicate that there were no statistically significant differences (*p* < 0.05) in any of the variables studied; therefore, it can be concluded that age was not a differential factor for these variables.

### 3.6. Correlation between OHRQoL and Anxiety Levels

Regarding the relationship between the OHRQoL and the anxiety levels of the patients, OHIP-14 and STAI scores were compared. The results are presented in Table 5.

It can be concluded that there was a correlation, mild in intensity (0.17) but significant (*p* < 0.05), between the dimension of physical disability from the OHIP-14 questionnaire and the STAI A/R. At this point, patients with a higher level of anxiety state/trait tended to describe a more negative impact on their OHRQoL in terms of physical disability and vice versa.

## 4. Discussion

The OHRQoL is a relative concept based on the patient’s own experience and perception; therefore, it is important to apply a validated and objective method to analyze the oral-health quality of life. The OHIP-14 questionnaire is one of the most widely used instruments to assess OHRQoL [25,27,28], especially in patients during their orthodontic treatment [29,30,31,32].

The level of anxiety in the patients undergoing orthodontic treatment has also been analyzed previously in different studies [22,23].

The sample used in this study was a homogeneous sample with respect to sex and age variables. No statistically significant differences were found in the study participants when analyzing for their sex and age.

When analyzing the OHRQoL in patients during their orthodontic treatment, the results described in this study were similar to those described by other authors previously [23,25,28,30,31,32]. This study concluded that the dimension with the highest score on the OHIP-14 questionnaire was psychological disability (3.20 ± 1.08) followed by functional limitation (2.52 ± 0.81), while handicap (0.37 ± 0.56) and social disability (0.43 ± 0.69) were the dimensions with the lowest impact on patients. We observed that the dimension of the OHRQoL of the OHIP-14 questionnaire with the highest score was psychological disability (3.20 ± 1.08). These data may indicate that there may be a relationship between the psychological disability and the anxiety of the patients in this study.

These results are consistent with those reported by other authors in this regard. Zheng et al. analyzed the OHRQoL in patients beginning their orthodontic treatment. They reported that after the cementing of the brackets, the dimension with the most negative impact on patients was also psychological disability (3.96 ± 1.28); social disability (0.74 ± 0.47) was the dimension with the lowest average score [31].

Kang et al. also evaluated the OHRQoL in patients undergoing orthodontic treatment. They concluded that handicap (1.54 ± 1.53) was the dimension in which patients described the lowest impact on their oral-health quality of life [32].

Machado et al. analyzed the OHRQoL using the OHIP-14 questionnaire one month after starting orthodontic treatment (as in this study) in patients with bruxism and without bruxism. These authors found the average values in the total questionnaire score to be 8.55–10.16 (±6.20–5.15). In this study, we observed that patients described similar values for the total score of the OHIP-14 questionnaire one month after starting their treatment (11.93 ± 2.19) [29].

In 2021, Gao et al. analyzed the OHRQoL using the OHIP-14 questionnaire comparing a group of adult patients with conventional brackets and another group of patients on a treatment with clear aligners. They concluded that the patients with clear aligners described less of an impact on their OHRQoL compared to the conventional bracket group during the initial phase of their orthodontic treatment. They also observed that patients with clear aligners described a lower level of anxiety compared to the bracket group [22].

Studies have been developed that evaluate the impact of orthodontic treatment on the OHRQoL of patients. These studies concluded that the OHIP-14 questionnaire scores were significantly lower after the receiving of orthodontic treatment, indicating an improvement in patients’ OHRQoL [25].

There are different studies that have evaluated dental anxiety in patients undergoing orthodontic treatment. It was observed that anxiety levels were higher on the second day of treatment and gradually decreased thereafter [33].

According to the scientific literature, when analyzing the influence of sex on anxiety levels in patients during their orthodontic treatment, there is some controversy within the different studies published. This study concluded that there were no statistically significant differences in anxiety levels when comparing for patient sex. These results agree with the results described by other authors [33,34]. In contrast, there are authors who have observed that women described higher levels of anxiety compared to men during their orthodontic treatment [5].

In our study, one month after the start of orthodontic treatment, we observed that patients described a mean dental anxiety score on the STAI questionnaire ranging from 27.93 to 26.78 (±2.92–2.91). Previous studies describe similar scores after a month of starting orthodontic treatment using fixed appliances; for example, Wang et al. reported an anxiety value of 31.0 ± 14.0 a month after the placing of an orthodontic arch. They observed that the degree of anxiety decreased from the first day of placement to reach that score at the one-month mark [23].

We found no statistically significant differences when assessing the influence of age on the anxiety levels of the patients evaluated. There are authors who observe that there is some influence from age on the anxiety levels in orthodontic patients. Yavan reported a positive correlation between age and the STAI-T score (*p* < 0.05), while he did not find a significant correlation between age and the STAI-S score. This author also concluded that sex influenced dental anxiety. The mean anxiety scores on the STAI questionnaire were higher for women than for men [35].

Carlsson et al. reported higher levels of anxiety in younger patients compared to older patients. They observed that there was a statistically significant relationship between the age of orthodontic patients and their degree of anxiety [36]; however, in our case we did not observe a significant difference when analyzing the influence from age on the level of anxiety.

This study concludes that sex does not have a statistically significant influence on the anxiety levels in adult patients undergoing orthodontic treatment. In the current scientific literature, there is controversy on this point. There are authors who also describe that sex does not significantly influence dental anxiety [29,37]. However, there are other studies that report higher levels of anxiety in female patients as compared to adult male patients [4,5,36].

This study focused on evaluating the impact of orthodontic treatment using fixed multibracket appliances on the OHRQoL. There are authors who have analyzed this impact by comparing a bracket system with a transparent aligner system. A systematic review by Zhang et al. concluded that, at present, based on the published scientific evidence, there are no results that can statistically significantly report that the use of one or another orthodontic system can produce significant improvements in the OHRQoL of patients [38].

Kumar et al. conducted a study in 2009 in which they analyzed, as in this study, the influence of anxiety levels on the patients’ OHRQoL. A total of 1235 individuals aged 15 to 45 years were analyzed using the Corah dental anxiety scale and the OHQoL-UK OHRQoL questionnaire. These authors concluded that women presented a higher level of dental anxiety compared to men. In relation to age, older individuals also presented a higher level of dental anxiety than younger patients. Regarding the results of our study, it can be justified by seeing that both studies used different scales to average the anxiety and OHRQoL of patients and did not evaluate participants in active orthodontic treatment. Kumar et al. observed that anxiety had a significant influence on the OHRQoL, describing people with high anxiety as having a worse OHRQoL [39]. They also concluded that older and female patients described higher levels of anxiety as compared to younger and male patients.

Anxiety can occur during different stages of orthodontic treatment. Patients who are about to undergo orthognathic surgery describe high levels of anxiety before undergoing surgery [40,41]. Barel et al. in 2018 published a study in which they concluded that the more information or knowledge patients had about the orthognathic surgery procedure, the lower the level of preoperative anxiety described by the patients. From the above, it can be concluded that it is necessary to inform patients correctly concerning the procedure in order to reduce their anxiety levels during orthodontic treatment [41].

Patients also describe anxiety during the debonding process of conventional brackets. Bracket removal should be a quick and painless procedure. The anxiety that patients may describe prior to bracket removal can lead to pain during the procedure. Orthodontists should inform the patient about how bracket removal will be performed to try to reduce the level of anxiety and pain [42].

Orthodontic treatment with intraoral and extraoral appliances also causes anxiety in child patients [43,44]. Topcuoglu et al. described how extraoral appliances have negative effects on patients and their parents’ anxiety levels. This author concluded that the advantages and disadvantages of extraoral appliances must be evaluated in order not to negatively affect the patient’s mood. Informing patients about possible problems and discomfort throughout functional treatment is beneficial in order to enhance both appliance efficiency and patient compliance [44].

Patient cooperation during orthodontic treatment and compliance with the prescribed guidelines are essential for effective treatment. Patient compliance depends on a number of factors. These factors include the degree of pain and anxiety patients describe during their treatment and their expectations of treatment. This pain has a negative effect on the patient’s desire to undergo orthodontic treatment and on their subsequent willingness to adhere to instructions. Knowledge about esthetic and functional expectations may help to predict the patient’s pain response to orthodontic treatment [45,46].

Facial and dental aesthetics, especially of the smile, is a fundamental factor that affects patients’ psychosocial well-being and influences their oral quality of life. The main concern of patients before starting orthodontic treatment is to improve the aesthetics of their smile [47].

The search for strategies to reduce or eliminate anxiety in patients may contribute to decreasing their perception of pain. In recent times, several studies have been published that have developed and evaluated different methods for reducing anxiety during treatment. These include the use of biofeedback therapy [48,49], sending information to the patient via text messages [6], or the use of aromatherapy [50].

Orthodontic specialists and healthcare professionals in general should be vigilant in establishing effective communication with patients and providing them with full information about their treatment. This increased care and contact with patients should create greater confidence between the orthodontist and their patients, which will lead to a decrease in the level of anxiety and stress the patients experience during their orthodontic treatment.

It has been shown that with in-person healthcare, patient interaction always plays a central role, but it is important not to forget the new remote systems that have been implemented in recent times as a good alternative to face-to-face care. Telephone calls and text messages, for example, during orthodontic treatment have been shown to be effective in reducing patients’ levels of anxiety and also in reducing pain perception [6].

The results described in this study indicate that patients undergoing orthodontic treatment will describe anxiety during their treatment. It is important for clinicians to be able to properly inform patients about possible complications or discomfort prior to treatment in order to try to reduce the level of anxiety that patients may feel. In addition, the orthodontic forces applied for tooth alignment also play an important role in promoting pain and thus generating anxiety. Anxiety is pointed to as the main psychological aspect related to the perception of pain. The increased perception of pain and anxiety may hamper social and daily activities, especially eating and sleeping.

Orthodontists must pay attention to the anxiety that patients present before and during their treatment in order to try to apply methodologies that can reduce anxiety. Based on the results, the present study suggests that it is important to explain in detail to patients before starting orthodontic treatment the possible discomfort they may experience in order to try to reduce their degree of anxiety during their treatment and to improve the subsequent impact on their oral quality of life.

A previous study evaluated the influence of conventional metal bracket slot size on patients’ pain levels and the impact on their OHRQoL. It was observed that the bracket slot size could influence how patients described the impact of their orthodontic treatment on their OHRQoL. A trend was reported in which patients with conventional brackets with a slot size of 0.018′′ described a higher level of pain and a more negative impact on their OHRQoL compared to patients with brackets with a slot size of 0.022′′. It would be interesting in future studies to analyze the influence of other variables (e.g., orthodontic archwire size, type of orthodontic archwire material, etc.) on anxiety and the OHRQoL of orthodontic patients [51], as well as to evaluate the influence of the orthodontic system used (conventional metal brackets, aesthetic brackets, self-ligating brackets or clear aligners) on patients’ anxiety levels and OHRQoL.

This study had some limitations that may hinder the generalization of the results to the general population. One of the main limitations was not being able to analyze a control group due to the nature of the study. In the absence of a control group for this study, it was not possible to determine a direct causal relationship based on the results described in this paper. Another limitation was not grouping patients according to the type of malocclusion they had. It is possible that, as well as potentially having a lesser or greater impact on their OHRQoL, depending on the type of malocclusion, patients may describe higher or lower levels of anxiety about the malocclusion. Another limitation of this study was the use of self-declared questionnaires by the participants. It would also be useful in future research to analyze the influence of the type of orthodontic treatment system (lingual brackets, self-ligating brackets, aesthetic brackets, and transparent aligners) on anxiety levels and OHRQoL. Because of this, it is of great relevance to continue this type of study using longitudinal designs. Future studies should be conducted on larger samples.

## 5. Conclusions

Patients with higher trait anxiety described a more negative impact on this dimension of their OHRQoL.

Psychological disability was the dimension of the OHIP-14 questionnaire that had the greatest impact on the OHRQoL. The dimension with the lowest impact in this study was disability.

No statistically significant influence was observed from analyzing age and sex on the anxiety levels and OHRQoL of patients one month after starting their orthodontic treatment.

When analyzing the influence of dental anxiety and OHRQoL in a sample of adult patients undergoing orthodontic treatment using conventional brackets, it can be concluded, based on the results of this study, that the physical disability dimension of the OHRQoL OHIP-14 questionnaire was associated with an increased level of anxiety in adult patients treated with conventional brackets. Although it could not be demonstrated in this study, the impact of orthodontic treatment on adult patients may negatively influence their anxiety levels.

## Figures and Tables

**Figure 1 ijerph-19-10767-f001:**
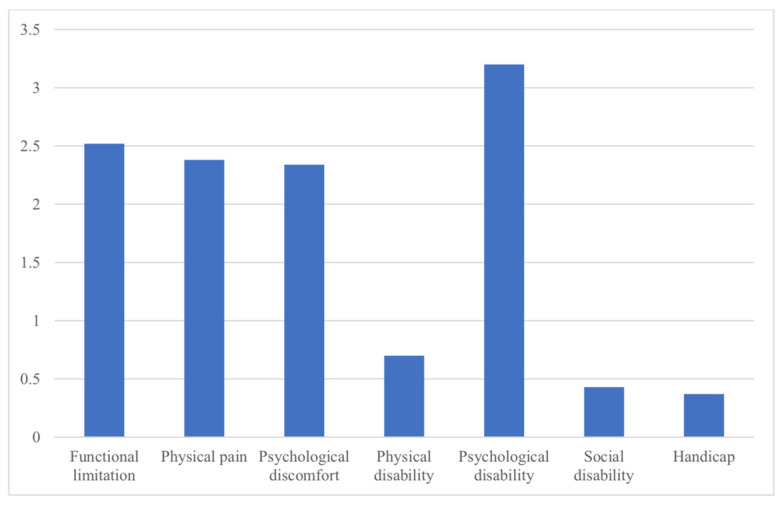
Dimensions of the OHIP-14 questionnaire.

**Table 1 ijerph-19-10767-t001:** Descriptive analysis. Variables in the STAI questionnaire (*n* = 120).

Variable	Exploration: Shape			Centrality		Range (Min./Max.)	Variability	
	Skewness	Kurtosis	KS Test: *p* Value	Mean [95% CI]	Medium		Standard Deviation	Interquartile Range
Anxiety State	−0.52	−0.55	0.008 **	27.93 (27.41–28.46)	28.00	21/32	2.92	4.75
Trait Anxiety	0.08	−0.47	0.305 ^NS^	26.78 (26.26–27.31)	27.00	21/34	2.91	4.00

^NS^ = No significant deviation (*p* > 0.05), and the variable is normally distributed. ** = Significant severe deviation (*p* < 0.01), and the variable does not adjust to normal. KS = Kolmogorov–Smirnov.

**Table 2 ijerph-19-10767-t002:** Descriptive analysis. OHIP-14 questionnaire variables (*n* = 120).

Domains	Exploration: Shape			Centrality		Range (Min./Max.)	Variability	
	Skewness	Kurtosis	KS Test: *p* Value	Mean [95% CI]	Medium		Standard Deviation	Interquartile Range
Functional limitation	0.72	0.47	0.000 **	2.52 (2.37–2.66)	2.00	0/5	0.81	1.00
Physical pain	1.02	−0.02	0.000 **	2.38 (2.28–2.47)	2.00	2/4	0.54	1.00
Psychological discomfort	1.29	1.11	0.000 **	2.34 (2.23–2.46)	2.00	1/4	0.64	1.00
Physical disability	1.05	0.44	0.000 **	0.70 (0.55–0.85)	0.50	0/3	0.84	1.00
Psychological disability	0.93	1.03	0.000 **	3.20 (3.00–3.40)	3.00	1/7	1.08	2.00
Social disability	1.50	1.39	0.000 **	0.43 (0.30–0.55)	0.00	0/3	0.69	1.00
Handicap	1.26	0.63	0.000 **	0.37 (0.26–0.47)	0.00	0/2	0.56	1.00
OHIP-14 Total	0.11	−0.48	0.196 ^NS^	11.93 (11.53–12.32)	12.00	8/17	2.19	3.75

^NS^ = No significant deviation (*p* > 0.05) and the variable is normally distributed. ** = Significant severe deviation (*p* < 0.01) and the variable does not adjust to normal. KS = Kolmogorov–Smirnov.

**Table 3 ijerph-19-10767-t003:** Inferential Analysis. Intergroup comparison of OHIP-14 and STAI variables based on sex (*n* = 120).

Variables	Median/Mean (S.D.)		Statistical Hypothesis Test		Effect Size: R^2^
	Men (*n* = 52)	Women (*n* = 68)	Value	*p*-Value	
OHIP—Functional Limitation	2.00/2.46 (0.73)	2.00/2.46 (0.73)	Z_U_ = 0.74 ^NS^	0.459	0.004
OHIP—Physical Pain	2.00/2.38 (0.49)	2.00/2.38 (0.49)	Z_U_ = 0.50 ^NS^	0.619	0.000
OHIP—Psychological Discomfort	2.00/2.37 (0.63)	2.00/2.37 (0.63)	Z_U_ = 0.68 ^NS^	0.500	0.001
OHIP-Physical Disability	0.00/0.60 (0.80)	0.00/0.60 (0.80)	Z_U_ = 1.24 ^NS^	0.213	0.012
OHIP—Psychological Disability	3.00/3.15 (1.07)	3.00/3.15 (1.07)	Z_U_ = 0.52 ^NS^	0.604	0.001
OHIP—Social Disability	0.00/0.46 (0.67)	0.00/0.46 (0.67)	Z_U_ = 0.84 ^NS^	0.398	0.002
OHIP—Handicap	0.00/0.44 (0.64)	0.00/0.44 (0.64)	Z_U_ = 1.01 ^NS^	0.311	0.014
OHIP—Total score	12.00/11.88 (2.27)	12.00/11.88 (2.27)	t = 0.28 ^NS^	0.776	0.001
STAI—Anxiety State A/E	28.00/27.60 (2.77)	28.00/27.60 (2.77)	Z_U_ = 1.37 ^NS^	0.170	0.010
STAI—Anxiety Trait A/R	26.50/26.54 (2.69)	26.50/26.54 (2.69)	t = 0.80 ^NS^	0.423	0.005

^NS^ = Not significant (*p* > 0.05).

**Table 4 ijerph-19-10767-t004:** Inferential Analysis. Intergroup comparison of OHIP-14 and STAI variables based on age (*n* = 120).

Variables	Median/Mean (S.D.)		Statistical Hypothesis Test		Effect Size: R^2^
	19–30 Years (*n* = 56)	31–45 Years Old (*n* = 64)	Value	*p*-Value	
OHIP—Functional Limitation	2.00/2.61 (0.73)	2.00/2.44 (0.71)	Z_U_ = 1.12 ^NS^	0.262	0.011
OHIP—Physical Pain	2.00/2.36 (0.49)	2.00/2.39 (0.52)	Z_U_ = 0.51 ^NS^	0.608	0.001
OHIP—Psychological Discomfort	2.00/2.29 (0.63)	2.00/2.39 (0.66)	Z_U_ = 0.74 ^NS^	0.458	0.007
OHIP-Physical Disability	0.50/0.70 (0.80)	0.50/0.70 (0.85)	Z_U_ = 0.00 ^NS^	0.998	0.000
OHIP—Psychological Disability	3.00/3.14 (1.07)	3.00/3.25 (1.07)	Z_U_ = 0.93 ^NS^	0.350	0.002
OHIP—Social Disability	0.00/0.30 (0.67)	0.00/0.53 (0.73)	Z_U_ = 1.93 ^NS^	0.053	0.027
OHIP—Handicap	0.00/0.34 (0.64)	0.00/0.39 (0.55)	Z_U_ = 0.73 ^NS^	0.466	0.002
OHIP—Total score	12.00/11.73 (2.27)	12.00/12.09 (2.06)	t = 0.80 ^NS^	0.369	0.007
STAI—Anxiety State A/E	28.00/28.16 (2.77)	28.00/27.73 (3.08)	Z_U_ = 0.63 ^NS^	0.528	0.005
STAI—Anxiety Trait A/R	27.00/26.71 (2.69)	27.00/26.84 (3.02)	t = 0.24 ^NS^	0.809	0.000

^NS^ = Not significant (*p* > 0.05).

**Table 5 ijerph-19-10767-t005:** Associative analysis between OHIP-14 and STAI variables (*n* = 120).

Variables	Spearman R Value (*p*-Value)	
	STAI A/E	STAI A/R
OHIP—Functional Limitation	0.15 (0.052)	0.09 (0.161)
OHIP—Physical Pain	0.03 (0.376)	0.01 (0.442)
OHIP—Psychological Discomfort	0.12 (0.095)	0.11 (0.112)
OHIP—Physical Disability	0.04 (0.330)	0.17 (0.031) *
OHIP—Psychological Disability	0.09 (0.157)	0.00 (0.499)
OHIP—Social Disability	0.02 (0.428)	0.06 (0.265)
OHIP—Handicap	0.05 (0.294)	0.01 (0.450)
OHIP—Total score	0.08 (0.204)	0.07 (0.220)

* = Significant (*p* < 0.05).

## Data Availability

The data presented in this study are available on request from the corresponding author.

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
