# Peer review of "Oral-Health-Related Quality of Life and Anxiety in Orthodontic Patients with Conventional Brackets"

_ijerph, 2022, doi:10.3390/ijerph191710767_

Round 1

Reviewer 1 Report

Dear Authors,

Congratulations on the methodology and language of your manuscript! However, I have the following considerations and suggestions:

1. Title: Remove the word "Covid-19". You did not test the impact of the pandemic in your study. You did not even discuss it.

2. In the abstract, you concluded that anxiety negatively influenced the perceived dental impact on self-esteem. You only found a statistically significant relationship between anxiety and physical disability, but it was not possible therein to establish a cause and effect relationship. In reality, if I were allowed to guess on this, I'd assume that the physical disability (ortho Tx) increased anxiety, not the other way around. 

3. Table 2: Please explain under the table what KS means

3. Discussion: Throughout the text, there are expressions such as "other authors" or Previous studies, all in the plural, but there is only one citation or even none. There was no discussion on what the clinical take-home message is. After reading your manuscript, how shall clinicians change the way they practice or advise their patients? We now know that braces cause discomfort and that might increase anxiety levels. So what? Don't you already know this? You even cited previous studies in which  aligners were perceived as more comfortable than braces. Then, for those already anxious patients,  should we prescribe aligners? What is your advice now after having all that work? There was no in-depth discussion about these things and, in the end of the day, these are all that matter, or am I missing something? 

The info presented may still be helpful in the bulk of the literature, but you need to be more convincing regarding its importance, and present it in a way that resonates with the results and the title instead of making assumptions that can't be supported. I hope this was helpful!

Reviewer 2 Report

The manuscript is interesting and well designed. However, the title is Oral-Health-Related Quality of Life and Anxiety in Orthodontic Patients during the COVID-19 Pandemic. I do not understand why you mention COVID-19. You do not mention the pandemic anywhere in the manuscript, and I also do not know what COVID-19 has to do with your topic.

Reviewer 3 Report

Dear Authors,

although I thank the paper is extremely interesting, I would like to suggestions you widen some topics and aspects in Your paper:

1. In the introduction section, I would like you to hightlighten

- the influence of patients cooperation on the orthodontic treatment:

Sarul M, Kawala B, Kozanecka A, Łyczek J, Antoszewska-Smith J. Objectively measured compliance during early orthodontic treatment: Do treatment needs have an impact? Adv Clin Exp Med. 2017 Jan-Feb;26(1):83-87. doi: 10.17219/acem/62107. PMID: 28397437.

(add to discussion)

Kozanecka A, Sarul M, Kawala B, Antoszewska-Smith J. Objectification of Orthodontic Treatment Needs: Does the Classification of Malocclusions or a History of Orthodontic Treatment Matter? Adv Clin Exp Med. 2016 Nov-Dec;25(6):1303-1312. doi: 10.17219/acem/62828. 

Sarul M, Lewandowska B, Kawala B, Kozanecka A, Antoszewska-Smith J. Objectively measured patient cooperation during early orthodontic treatment: Does psychology have an impact? Adv Clin Exp Med. 2017 Nov;26(8):1245-1251. doi: 10.17219/acem/65659. PMID: 29264882.

- the need for the esthetic, symetric smile, especially in the arterior region:

Chrapla, P.; Paradowska-Stolarz, A.; Skoskiewicz-Malinowska, K. Subjective and Objective Evaluation of the Symmetry of Maxillary Incisors among Residents of Southwest Poland. Symmetry 2022141257. https://doi.org/10.3390/sym14061257

2. Please, add the information weather or not the OHIP-14 needed to be validated for the Salamanka studies

3. In the discussion, please add the aspect of dental anxiety among the patients treated art the pandemic, eg. 

Berberoğlu B, Koç N, Boyacioglu H, et al. Assessment of dental anxiety levels among dental emergency patients during the COVID-19 pandemic through the Modified Dental Anxiety Scale. Dent Med Probl. 2021;58(4):425–432. doi:10.17219/dmp/139042

4. In the discussion section, it would be valid to add the information on the differences of the shear bond strenght to the different surfaces and compare it to the forces use during treatment (może frequent debondig etc), eg.

Shirazi M, Mirzadeh M, Modirrousta M, Arab S. Comparative evaluation of the shear bond strength of ceramic brackets of three different base designs bonded to amalgam and composite restorations with different surface treatment. Dent Med Probl. 2021;58(2):193–200. doi:10.17219/dmp/131684

The aspect of more restaurations might be crucial in the fear level, because the patient probably has more  „bad” experiences in the treatment, but on the other hand, debonded brackets may cause force distortions and cause less „pain”

5. In the discussion, it would be albo nice to add some information on „other” appliances, eg. Hyrax (there are several Articles refering to the anxiety while this appliance is used)

6.I also think that the interesting aspect in the dental anxiety might be a biofeedback, that is rather a different „procedure”, that to my mind reduces dental anxiety - please, note:

Florjanski W, Malysa A, Orzeszek S, Smardz J, Olchowy A, Paradowska-Stolarz A, Wieckiewicz M. Evaluation of Biofeedback Usefulness in Masticatory Muscle Activity Management-A Systematic Review. J Clin Med. 2019 May 30;8(6):766. doi: 10.3390/jcm8060766. 

7. In the limitations, I would also add preparatom for orhognatic surgery, especially SARPE = did Patients have this mind of procedures? The orthognathic surgery is usually some aspect that patients fear for, do this may also influence the aspect of anxiety in general. Also, were there any information on the previous „painful” dental treatment?

Round 2

Reviewer 1 Report

This paper suffices to report an association between physical disability and anxiety, but it does not prove a cause and effect relationship. The Spearman test is a correlation test that explains the strength and direction of an association. Even though the discussion was improved with the addition of studies supporting that physical disability might cause anxiety, your Spearman test result does not support to what you wrote in the conclusion. You wrote: "Conclusions: The physical disability dimension of the OHIP14 questionnaire increased the anxiety level of adult patients treated with conventional brackets. The impact of orthodontic treatment on adult patients may can negatively influence their level of anxiety".

You should have written: "Conclusions: The physical disability dimension of the OHIP14 questionnaire was associated with an increase in the anxiety level of adult patients treated with conventional brackets.  Although it was not possible to prove in this study, the impact of orthodontic treatment on adult patients may negatively influence their level of anxiety."

The bottom line is that although there is an association, you cannot determine if braces were indeed the culprit. As the study seems to have been conducted during the pandemic, and you did not have a control group, the anxiety could have been due to the pandemic or even something else. 

Your study is a reasonable one. Good enough per se, with no need to prove that one thing leads to the other. It suffices to say that the association exists and that you suspect of what the cause may be but, due to the absence of a control group, and the existence of other lurking variables, a cause and effect relationship can't be determined by this paper. This is on what your discussion and conclusion should be focused, in my humble opinion. 

Reviewer 3 Report

Dear Authors,

Thank you for all the explanations, especially concerning OHIP - validation.

I think it would be valid to add your paper to the discussion to show that it is a planned and continuous research, although I understand the concerns with self-citations. Please, consider that - that is the only reason I give you "minor revision" at this stage of work.

eg.
Curto A, Albaladejo A, Montero J, Alvarado-Lorenzo M, Garcovich D, Alvarado-Lorenzo A. A
Prospective Randomized Clinical Trial to Evaluate the Slot Size on Pain and Oral Health-Related
Quality of Life (OHRQoL) in Orthodontics during the First Month of Treatment with Conventional
and Low-Friction Brackets. Applied Sciences. 2020; 10(20):7136

I would like to congratulate you on the paper you wrote and suggest further studies in that area.
